**Data Availability Statement:** All relevant data are within the manuscript.

**Funding:** The authors received no specific funding for this work.

# Prevalence and associated factors of impaired renal function and albuminuria among adult patients admitted to a hospital in Northeast Ethiopia

**Temesgen Fiseha**[1]*, **Ermiyas Ahmed**[1], **Semagn Chalie**[1], **Angesom Gebreweld**[2]

**1** Department of Clinical Laboratory Science, College of Medicine and Health Sciences, Wollo University, Dessie, Ethiopia, **2** Department of Medical Laboratory Science, College of Health Sciences, Mekelle University, Mekelle, Ethiopia

\* temafiseha@gmail.com

## Abstract

### Background

Chronic kidney disease (CKD) is increasingly common in hospitalized patients and is associated with increased risk for in-hospital morbidity and mortality. However, data regarding the prevalence of CKD in the African hospitalized patient population are limited. We therefore examined the prevalence and associated factors of impaired renal function and albuminuria among adult patients admitted to the internal medicine wards of a hospital in Northeast Ethiopia.

### Methods

A cross-sectional study was conducted from January 1 to April 30, 2020 at the inpatient settings of Dessie referral hospital. Data on demographics and medical history were obtained, and serum creatinine and albuminuria were analyzed. Estimated glomerular filtration rate (eGFR) was calculated using the Modification of Diet in Renal Disease (MDRD) equation. CKD was defined as impaired eGFR ($<60$ ml/min/$1.73m^2$) and/or albuminuria. Univariate and multivariable analysis were conducted to determine factors associated with impaired eGFR and albuminuria.

### Results

A total of 369 patients were included in this study. The prevalence of impaired eGFR was 19.0% (95%CI: 15.2%–23.2%) and albuminuria was 30.9% (95%CI: 26.3%–35.7%). Overall, 33.9% (95%CI: 29.2%–38.9%) of the patients had some degree of CKD, but only 21.6% (95%CI: 15.1%–29.4%) were aware of their renal disease. In multivariable analysis, older age, a family history of kidney disease, diabetes, hypertension and HIV were independently associated with both impaired eGFR and albuminuria while male gender was independently associated with only albuminuria.

**Competing interests:** The authors have declared that no competing interests exist.

**Abbreviations:** AOR, Adjusted odds ratio; BP, Blood pressure; CKD, Chronic kidney disease; CI, Confidence interval; DRH, Dessie Referral hospital; eGFR, Estimated glomerular filtration rate.

## Conclusions

CKD is common in adult patients admitted to the internal medicine wards, but only few patients are aware of their condition. These findings highlight the need for feasible approaches to timely identify kidney disease and raise awareness on the importance of detection and early intervention in the inpatient settings.

## Introduction

Chronic kidney disease (CKD), typically defined by impaired estimated glomerular filtration rate (eGFR) and albuminuria, is a major global public health problem [1]. CKD affects 11 to 13% of the population worldwide and is an independent risk factor for cardiovascular and all-cause mortality [2, 3]. It has been also shown to be a risk factor for cardiovascular disease and is associated with adverse outcomes, including hospitalizations and progression to kidney failure, which have enormous impacts on the quality of life and health care system [4, 5]. Early detection, intervention and management of patients with CKD is therefore crucial to reduce the morbidity and mortality, delay the progression of disease and improve health outcomes.

CKD is increasingly common in hospital inpatient settings, occurring in up to 39% of hospitalized patients [6]. In hospitalized patients, CKD has been found to be associated with an increased risk for length of hospital stay, acute renal failure, in-hospital mortality and health-related expenditure [7–10]. The presence of CKD has been also shown to be a risk factor for adverse outcomes, including drug toxicity, dose adjustment issues, infections and poor functional status [11–14]. Hospitalized patients with CKD also have a substantial burden of comorbidities, including underlying diseases and consequences of CKD such as diabetes, hypertension, anemia, and bone and mineral disease, that contribute to increase the risk for adverse outcomes and makes the management of these patients potentially challenging [15–17]. Early detection is therefore desirable because effective interventions can then be implemented to reduce the risk of in-hospital morbidity and mortality associated with CKD, and improve outcomes.

Although hospitalization represents an opportunity to identify existing CKD and educate patients earlier in the course of their kidney disease, CKD is often unrecognized and most patients with kidney disease are unaware of their condition [18–21]. Furthermore, the attention paid to this condition is poor [18, 22]. However, despite the high prevalence of CKD and its resulting increased in-hospital morbidity and mortality, little is known about the prevalence of CKD in the African hospitalized patients. The few available studies among hospitalized patients in Africa have reported CKD prevalence of 13.5% in Botswana [23], 38.6% in Kenya [6] and 57.3% in Uganda [24]. Understanding the burden and associated risk factors of CKD based on relevant indicators of kidney disease is important for making relevant decisions regarding identification and prevention of the disease in this resource limited region, where access to renal replacement therapy is strictly rationed [25]. We therefore examined the prevalence and awareness of CKD and identified the factors associated with impaired eGFR and albuminuria among adult patients admitted to the internal medicine wards of a hospital in Northeast Ethiopia.

## Methods

### Study design, setting and population

This cross-sectional study was conducted from January 1 to April 30, 2020 at the inpatient settings of Dessie referral hospital (DRH), Northeast Ethiopia. DRH is a 597 bed medical center, and serves as a tertiary referral hospital for South Wollo and surrounding Zones of Amhara

regional state. The hospital provides 350,000 outpatient and 23,780 inpatient services annually. Patients were eligible for the study if they were aged 18 years or older, were admitted to the internal medicine wards for at least 48 hours, and had serum creatinine measurements at admission. Patients admitted to intensive care units, patients with possibility of functional proteinuria and patients who had evidence of factors that can cause acute kidney injury or those on medical diagnosis of renal failure were excluded. A total of 369 patients who fulfilled the above criteria were consecutively included for the final analysis. The sample size was calculated on the basis of the following assumptions: use of the single proportion formula [26] with a 95% confidence level; 5% margin of error; 32.7% prevalence of impaired eGFR [27]; and 10% non-response rate. The study was approved by the Institutional Review Board of College of Medicine and Health Sciences, Wollo University (# 135/13/12). Written informed consent was obtained from each study participants after explaining the purpose and procedures of the study. Clinical information obtained in this study was communicated to attending physicians so that they could be used for clinical care.

## Data collection and measurements

Data were collected from patients and their medical records using structured questionnaire which was developed in English with modification from Screening and Early Evaluation of Kidney Disease (SEEK) study [28]. The questionnaire was carefully designed and pre-tested on 5% of study population and, based on the results, revision was made to minimize errors. Patients were interviewed to collect data on socio-demographic characteristics, family history of kidney disease and lifestyle behaviors. Trained and certified personnel abstracted data on comorbidities, including diabetes mellitus, hypertension, cardiovascular disease (coronary artery disease, myocardial infarction, heart failure, peripheral vascular disease, and old stroke), diseases of the respiratory system, HIV/AIDS and all others from admission medical records. All comorbidities were defined as present if documented in the medical records. Blood pressure was measured with a mercury sphygmomanometer after the patients had rested for 5–10 min in the sitting position. Three readings were taken 5 min apart, and the mean of these readings was recorded. A fasting venous blood sample and spot urine specimen were collected from each patient in the morning and then transported to the hospital inpatient laboratory. Serum creatinine was measured using the Jaffe kinetic method with calibration traceable to isotope dilution mass spectrometry (IDMS). Estimated glomerular filtration rate (eGFR) was calculated using the Modification of Diet in Renal Disease (MDRD) equation [29]. Impaired renal function was defined as eGFR $<60$ ml/min/1.73 m$^2$. Albuminuria was determined using rapid test strips (COMBINA 11S, Human) and was defined as a dipstick of $\geq 1+$. All laboratory measurements were done following the standard procedures recommended by the manufacturer. CKD was defined as the presence of impaired eGFR ($<60$ ml/min/1.73 m$^2$) and/or albuminuria. The CKD stages were classified according to the NKF Kidney Disease Outcomes Quality Initiative (KDOQI) guideline [30]: stage 1, eGFR $\geq 90$ ml/min/1.73 m$^2$ with albuminuria; stage 2, eGFR 60–89.9 ml/min/1.73 m$^2$ with albuminuria, and stages 3, 4 and 5 as eGFR 30–59.9, 15–29.9 and $< 15$ ml/min/1.73 m$^2$, respectively. Stage 3 was further classified into 3a (45–59.9 ml/min/1.73 m$^2$) and 3b (30–44.9 ml/min/1.73 m$^2$) [31]. CKD awareness among patients with kidney disease was defined as a 'yes' response the question "Has a doctor or other health care provider ever told you that you have a failing kidney or kidney disease (excluding kidney stones, bladder infections, or incontinence)?".

## Statistical analysis

Data collected were entered into EpiData version 3.1 software (Epidata Association, Odense, Denmark) and analyzed using SPSS version 20 software (SPSS Inc., Chicago, IL, USA). We

derived means for continuous variables and proportions to describe the characteristics of the study patients as well as the prevalence of impaired renal function, albuminuria and CKD. Comparisons of patients according to the presence of impaired renal function or albuminuria were performed using Chi-square ($x^2$) test and t-test, where appropriate. To determine which factors were associated with the presence of impaired renal function or albuminuria, univariate analysis was conducted with age, sex, residence, education, smoking status, family history of kidney disease, presence of diabetes, hypertension, cardiovascular diseases, respiratory diseases and HIV/AIDS, and systolic and diastolic BP as variables. Variables that were found to be significant in univariate analysis ($P < 0.25$) were included in the multivariable backwards stepwise logistic regression model to identify factors independently associated with impaired renal function or albuminuria. $P$-value $< 0.05$ was used to indicate statistical significance.

## Results

### Demographic and clinical characteristics of the study patients

A total of 369 patients admitted to the internal medicine wards were included in this study. Table 1 shows the demographic and clinical characteristics of the study patients. The mean (± SD) age was 48.8 ± 17.9 years, and 192 (52.0%) were females. Most of the patients were below 60 years of age (65.0%), and 35.0% were ≥ 60 years old. Two hundred and thirty-nine (64.8%) were rural residents and 198 (53.7%) had no formal education. About 8% (7.9%) of patients

**Table 1. Demographic and clinical characteristics of the study patients (n = 369).**

| Characteristics | Category | |
|---|---|---|
| Age (year), mean ± SD | | 48.8 ± 17.9 |
| Age group, n (%) | 18–39 years | 119 (32.2) |
| | 40–59 years | 121 (32.8) |
| | ≥ 60 years | 129 (35.0) |
| Sex, n (%) | Male | 177 (48.0) |
| | Female | 192 (52.0) |
| Residence, n (%) | Urban | 130 (35.2) |
| | Rural | 239 (64.8) |
| Education, n (%) | No formal education | 198 (53.7) |
| | Grade 1–8 | 91 (24.7) |
| | Grade 9–12 | 52 (14.1) |
| | College & above | 28 (7.6) |
| Smoking, n (%) | Yes | 29 (7.9) |
| | No | 340 (92.1) |
| HIV status, n (%) | Yes | 42 (11.4) |
| | No | 327 (88.6) |
| History of DM, n (%) | Yes | 93 (25.2) |
| | No | 277 (74.8) |
| History of HTN, n (%) | Yes | 92 (24.9) |
| | No | 278 (75.1) |
| History of CVD, n (%) | Yes | 83 (22.5) |
| | No | 286 (77.5) |
| Systolic BP (mmHg), mean ± SD | | 126.6 ± 22.7 |
| Diastolic BP (mmHg), mean ± SD | | 79.1 ± 12.3 |
| Serum Creatinine (mg/dl), mean ± SD | | 1.22 ± 0.7 |
| eGFR (ml/min/1.73 m$^2$), mean ± SD | | 86.7 ± 39.8 |

were current smokers and about 11% (11.1%) were HIV patients. The main clinical diagnosis for admission was diabetes mellitus (25.2%), followed by hypertension (24.9%) and cardiovascular diseases (22.5%). The mean systolic and diastolic blood pressure (BP) of the patients were 126.6 ± 22.7 and 79.1 ± 12.3 mm Hg, respectively. The mean serum creatinine level was 1.22 ± 0.7 mg/dl, and 51 (13.8%) patients had values > 1.5 mg/dl. The mean eGFR was 86.7 ± 39.8 ml/min/ 1.73 m$^2$.

## Prevalence of indicators of kidney disease

The prevalence of impaired eGFR ($< 60$ ml/min/1.73 m$^2$) was 19.0% (95% CI: 15.2%–23.2%) and that of albuminuria was 30.9% (95% CI: 26.3%–35.7%). The prevalence of both impaired eGFR and albuminuria increased with age ($P < 0.001$; Fig 1), and were significantly higher in the older patients (age $\geq$60 years; 37.2% and 45.0%) than in the younger adults (9.2% and 23.3%, respectively; both $P < 0.001$). The prevalence of impaired eGFR was not different between men (18.1%) and women (19.8%; $P = 0.675$), while that of albuminuria was highest in men than women (37.9% vs. 24.5%; $P = 0.005$).

## Prevalence and awareness of CKD

Overall, 33.9% (95% CI: 29.2%–38.9%) of the patients had some degree of CKD, i.e. they had either a significantly impaired eGFR ($<$,60 ml/min/1.73 m$^2$) and/or albuminuria. Classified by disease stage, 11 (3.0%) patients had stage 1, 44 (11.9%) had stage 2, 35 (9.5%) had stage 3a, 15 (4.1%) had stage 3b, 14 (3.8%) had stage 4 and 6 (1.6%) had stage 5 CKD. It was noted that none of the patients at stage 5 CKD received dialysis during their hospitalization. There was a graded trend for increasing CKD prevalence with advancing age ($P < 0.001$; Fig 1). The prevalence of CKD was highest among older patients (50.4%) compared with younger patients (25.0%, $P < 0.001$), and it was different in men and women (39.5% vs. 28.6%; $P = 0.027$).

Awareness of CKD was 21.6% (95% CI: 15.1%–29.4%) among patients found to have any degree of kidney disease (i.e., impaired eGFR and/or albuminuria). Awareness of CKD vary significantly by disease stage, which was 9.1%, 11.4%, 17.1%, 33.3%, 42.9% and 66.7% among patients with stage 1, stage 2, stage 3a, stage 3b, stage 4 and stage 5 CKD, respectively ($P = 0.005$) (Fig 2). Of 70 patients with impaired eGFR ($< 60$ ml/min/ 1.73 m$^2$; stages 3–5 of CKD), 30.0% reported having kidney disease. Awareness of kidney disease was 23.7% among all the patients with albuminuria.

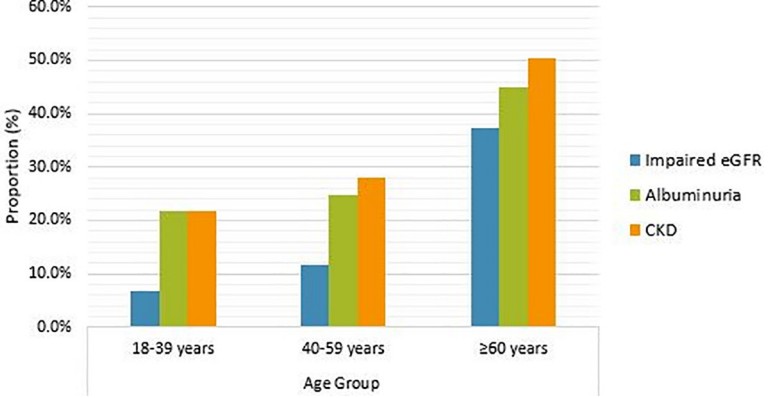

**Fig 1. Prevalence of impaired eGFR, albuminuria and CKD according to age group of study patients.**
Abbreviations: eGFR, estimated glomerular filtration rate; CKD, chronic kidney disease.

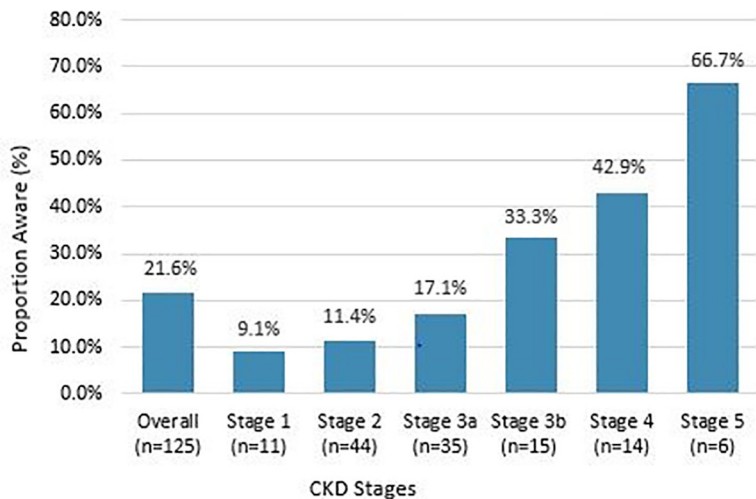

**Fig 2. Awareness of CKD stratified by CKD stages.** Abbreviations: CKD, chronic kidney disease.

### Factors associated with impaired renal function and albuminuria

Univariate and multivariable analysis were conducted to explore the factors associated with impaired renal function and albuminuria. On univariate analysis older age (OR = 5.87; 95% CI: 3.34–10.33), a family history of kidney disease (COR = 2.59; 95% CI: 1.34–4.92), history of diabetes (COR = 2.64; 95% CI: 1.53–4.50), hypertension (COR = 2.09; 95% CI: 1.20–3.64), respiratory disease (COR = 0.37; 95% CI: 0.14–0.95), HIV (COR = 2.75; 95% CI: 1.37–5.50) and systolic blood pressure (COR = 1.02, 95% CI: 1.00–1.04) were associated with impaired eGFR. In the multivariable analysis, older age (AOR = 6.42; 95% CI: 3.36–12.20), a family history of kidney disease (AOR = 3.08; 95% CI: 1.39–6.79), diabetes (AOR = 2.91; 95% CI: 1.41–6.00), hypertension (AOR = 3.83; 95% CI: 1.80–8.18) and HIV (AOR = 2.65; 95% CI: 1.15–6.09) remained independently associated with impaired eGFR (Table 2).

Using univariate analysis, we observed that older age (COR = 2.69; 95% CI: 1.70–4.24), male gender (COR = 1.88; 95% CI: 1.20–2.94), residence (COR = 1.62; 95%CI: 1.03–2.55), a family history of kidney disease (COR = 2.58; 95% CI: 1.41–4.74), diabetes (COR = 2.19; 95% CI: 1.35–3.58), hypertension (COR = 2.25; 95% CI: 1.38–3.67), respiratory disease (COR = 0.49; 95% CI: 0.24–0.98), HIV (COR = 2.85; 95% CI: 1.46–5.39) and systolic blood pressure (COR = 1.02, 95% CI: 1.00–1.03) were associated with albuminuria. In multivariable analysis, older age (AOR = 2.57; 95% CI: 1.51–4.40), male gender (AOR = 1.71; 95% CI: 1.02–2.87), family history of kidney disease (AOR = 2.63; 95% CI: 1.35–5.14), diabetes (AOR = 2.97; 95% CI: 1.65–5.35), hypertension (AOR = 3.60; 95% CI: 1.98–6.54) and HIV (AOR = 3.05; 95% CI: 1.42–6.56) were independently associated with the presence of albuminuria (Table 3).

### Discussion

The present study aimed to determine the prevalence and factors associated with impaired renal function (eGFR) and albuminuria in adult patients admitted to a hospital in Northeast Ethiopia. Findings from this study indicate that 19.0% of the patients had impaired eGFR ($< 60$ mL/min/1.73 m$^2$). A recent study conducted in adult patients admitted to Jimma University Medical Center in Southwest Ethiopia reported the prevalence of impaired eGFR to be 19.2% by the same definition with MDRD equation [27]. The prevalence of reduced eGFR

**Table 2. Factors associated with impaired eGFR among patients admitted to internal medicine wards of Dessie referral hospital, Northeast Ethiopia, 2020.**

| Characteristics | Impaired eGFR | | COR (95% CI) | P-value | AOR (95% CI) | P-value |
|---|---|---|---|---|---|---|
| | Yes (n = 70) | No (n = 229) | | | | |
| Age (years) | | | | < 0.001 | | < 0.001 |
| ≥ 60 | 48 (37.2) | 81 (62.8) | 5.87 (3.34–10.33) | | 6.42 (3.36–12.20) | |
| < 60 | 22 (9.2) | 218 (90.8) | 1 | | 1 | |
| Sex | | | | 0.675 | | |
| Female | 38 (19.8) | 154 (80.2) | 1.12 (0.66–1.86) | | NA | |
| Male | 32 (18.1) | 145 (81.9) | 1 | | NA | |
| Residence | | | | 0.854 | | |
| Rural | 46 (19.2) | 193 (80.8) | 1.05 (0.61–1.82) | | NA | |
| Urban | 11 (18.5) | 106 (81.5) | 1 | | NA | |
| Education | | | | 0.178 | | 0.807 |
| < High school | 59 (20.4) | 230 (79.6) | 1.61 (0.80–3.23) | | 1.11 (0.48–2.58) | |
| ≥ High school | 115 (13.8) | 69 (86.2) | 1 | | 1 | |
| Family history of kidney disease | | | | 0.004 | | 0.005 |
| Yes | 17 (34.0) | 33 (66.0) | 2.59 (1.34–4.92) | | 3.08 (1.39–6.79) | |
| No | 53 (16.6) | 266 (83.4) | 1 | | 1 | |
| Smoking | | | | 0.460 | | |
| Yes | 7 (24.1) | 22 (75.9) | 1.40 (0.57–3.42) | | NA | |
| No | 63 (18.5) | 277 (81.5) | 1 | | NA | |
| Diabetes mellitus | | | | 0.001 | | 0.004 |
| Yes | 29 (31.2) | 64 (68.8) | 2.64 (1.53–4.50) | | 2.91 (1.41–6.00) | |
| No | 41 (14.9) | 235 (85.1) | 1 | | 1 | |
| Hypertension | | | | 0.009 | | 0.001 |
| Yes | 26 (28.3) | 66 (71.7) | 2.09 (1.20–3.64) | | 3.83 (1.80–8.18) | |
| No | 44 (15.9) | 233 (84.1) | 1 | | 1 | |
| Cardiovascular disease | | | | 0.690 | | |
| Yes | 17 (20.5) | 66 (79.5) | 1.13 (0.62–2.09) | | NA | |
| No | 53 (18.3) | 233 (81.5) | 1 | | NA | |
| Respiratory disease | | | | 0.033 | | 0.371 |
| Yes | 5 (8.8) | 52 (91.2) | 0.37 (0.14–0.95) | | 0.57 (0.19–1.75) | |
| No | 65 (20.8) | 247 (79.2) | 1 | | 1 | |
| HIV | | | | 0.003 | | 0.022 |
| Yes | 15 (35.7) | 27 (64.3) | 2.75 (1.37–5.50) | | 2.65 (1.15–6.09) | |
| No | 55 (16.8) | 272 (83.2) | 1 | | 1 | |
| Systolic BP (mmHg) | 130.5 ± 21.5 | 125.7 ± 22.9 | 1.02 (1.00–1.04) | 0.040 | 1.01 (0.99–1.02) | 0.325 |
| Diastolic BP (mmHg) | 78.9 ± 9.8 | 79.2 ± 12.8 | 0.97 (0.95–1.00) | 0.089 | 0.98 (0.96–1.02) | 0.330 |

among patients admitted to a general medical ward in Uganda was found to be 15.3% by using the MDRD equation [24]. In Botswana, the prevalence of CKD stages 3–5 (eGFR $_{MDRD}$ <60 ml/min/1.73 m$^2$) among patients admitted to the medical wards was estimated at 16.3% [32]. Furthermore, in a retrospective cohort study of acute medical admissions in London, UK, the prevalence of renal impairment of the same degree using the same MDRD equation was found to be 17.7% [33]. The prevalence of albuminuria in this study was 30.9%, which was significantly higher than the prevalences reported from China [34] and Southwest Ethiopia [27], where 8.87% and 12.3% of the patients admitted to the medical wards had albuminuria with dipstick proteinuria of ≥ 1+, respectively. This greater prevalence in our study may reflect the more elderly and complex medical inpatient population.

**Table 3. Factors associated with albuminuria among patients admitted to internal medicine wards of Dessie referral hospital, Northeast Ethiopia, 2020.**

| Characteristics | Albuminuria | | COR (95% CI) | *P-value* | AOR (95% CI) | *P-value* |
|---|---|---|---|---|---|---|
| | Yes (n = 114) | No (n = 229) | | | | |
| Age (years) | | | | < 0.001 | | 0.001 |
| ≥ 60 | 58 (45.0) | 71 (55.0) | 2.69 (1.70–4.24) | | 2.57 (1.51–4.40) | |
| < 60 | 56 (23.3) | 184 (76.7) | 1 | | 1 | |
| Sex | | | | 0.005 | | 0.044 |
| Male | 67 (37.9) | 110 (62.1) | 1.88 (1.20–2.94) | | 1.71 (1.02–2.87) | |
| Female | 47 (24.5) | 145 (75.5) | 1 | | 1 | |
| Residence | | | | 0.037 | | 0.102 |
| Urban | 49 (37.7) | 81 (62.3) | 1.62 (1.03–2.55) | | 1.57 (0.92–2.69) | |
| Rural | 65 (27.2) | 174 (72.8) | 1 | | 1 | |
| Education | | | | 0.725 | | |
| < High school | 88 (30.4) | 201 (69.6) | 0.91 (0.54–1.55) | | NA | |
| ≥ High school | 26 (32.5) | 54 (67.5) | 1 | | NA | |
| Family history of kidney disease | | | | 0.002 | | 0.005 |
| Yes | 25 (50.0) | 25 (50.0) | 2.58 (1.41–4.75) | | 2.63 (1.35–5.14) | |
| No | 89 (27.9) | 230 (72.1) | 1 | | 1 | |
| Smoking | | | | 0.091 | | 0.064 |
| Yes | 13 (44.8) | 16 (55.2) | 1.92 (0.89–4.74) | | 2.64 (0.94–7.38) | |
| No | 101 (29.7) | 239 (70.3) | 1 | | 1 | |
| Diabetes mellitus | | | | 0.001 | | <0.001 |
| Yes | 41 (44.1) | 52 (55.9) | 2.19 (1.35–3.58) | | 2.97 (1.65–5.35) | |
| No | 73 (26.4) | 203 (85.1) | 1 | | 1 | |
| Hypertension | | | | 0.001 | | 0.001 |
| Yes | 41 (44.6) | 51 (55.4) | 2.25 (1.38–3.67) | | 3.60 (1.98–6.54) | |
| No | 73 (26.4) | 204 (73.6) | 1 | | 1 | |
| Cardiovascular disease | | | | 0.086 | | 0.138 |
| Yes | 32 (38.6) | 51 (61.4) | 1.56 (0.94–2.60) | | 2.40 (0.76–7.62) | |
| No | 82 (28.7) | 204 (71.3) | 1 | | 1 | |
| Respiratory disease | | | | 0.039 | | 0.210 |
| Yes | 11 (19.3) | 46 (80.7) | 0.49 (0.24–0.98) | | 2.06 (0.67–6.40) | |
| No | 103 (33.0) | 209 (67.0) | 1 | | 1 | |
| HIV | | | | 0.001 | | 0.022 |
| Yes | 22 (54.2) | 20 (47.6) | 2.81 (1.46–5.39) | | 3.05 (1.42–6.56) | |
| No | 92 (28.1) | 235 (71.9) | 1 | | 1 | |
| Systolic BP (mmHg) | 130.7 ± 22.2 | 124.8 ± 22.7 | 1.02 (1.00–1.03) | 0.030 | 1.01 (0.99–1.03) | 0.175 |
| Diastolic BP (mmHg) | 80.1 ± 11.8 | 78.7 ± 12.5 | 0.99 (0.97–1.02) | 0.451 | NA | |

About 34% (33.9%) of our patients admitted to the internal medicine wards had some degree of CKD (i.e., they had either impaired eGFR and/or albuminuria). This is comparable to the Conakry study, in which 33% of patients admitted to the medical wards had CKD with markers of kidney damage (proteinuria, hematuria, pyuria, renal morphological abnormalities) and impaired eGFR calculated using the MDRD formula [35]. In the Chinese retrospective cross-sectional study, 14.82% of hospitalized adult patients had CKD using the same indicators of renal disease [34]. The Brazilian retrospective study similarly reported a lower prevalence of CKD of 12.7% among adult patients admitted to the internal medicine wards, but the CKD criterion was based on the presence of medical diagnosis in medical records [36]. Other studies conducted in Kenya [6] and Uganda [24] reported a higher prevalence of CKD

compared to our study: 38.6% and 57.3%, respectively. The discrepancy could be explained by the differences in CKD definition used as well as methods for assessing albuminuria and eGFR. In the Kenyan study, for example, a diagnosis of CKD was defined as presence of markers of renal damage (including renal imaging, serum phosphate and calcium levels) and eGFR as determined by the Chronic Kidney Disease Epidemiology collaboration equation. The Uganda study, on the other hand, included microalbuminuria from Urine Albumin: Creatinine Ratio formula, renal imaging and eGFR as a definition criterion for CKD. The differences could also be due to variation in the lifestyle and age distribution of the studied patients.

Our study shows a lower awareness of CKD among the patients admitted to the internal medicine wards, only 21.6% of the adults with any degree of kidney disease (impaired eGFR and/or albuminuria) were aware of their condition. Although awareness was higher among patients with advanced disease, even among those with CKD stage 3b, awareness was only 33.3%, less than 50% for stage 4 and 66.7% with stage 5. Similar results have been reported in some of previous studies on hospital inpatients. In a retrospective study of general medicine inpatients from the University of Chicago Hospitalist Project, only 32% of patients with CKD were aware of their CKD. In addition, only 48% of patients with CKD stage 4 and 63% with stage 5 were aware of their disease [21]. In a cross-sectional study of general medicine inpatients at an urban academic medical center, awareness of CKD was at 33% [37]. In the Belgium study, more than a third of the CKD patients were not aware of their condition and only 65% of those with CKD stage 3b or 4 were aware of suffering from renal failure [22]. In the Botswana study, over half (53.5%) of the CKD cases were unaware of their disease [32].

The high prevalence and low awareness of CKD in this study support the evidence that CKD is frequently unrecognized in the inpatient settings, and that awareness is low among both physicians and affected patients [18–20]. In in-hospital patients, CKD is often not recognized until it is advanced and poorly documented in the medical record despite being present [19, 34, 38]. Even in CKD stage 3a or higher, it has been reported that as many as 70% of patients don't carry a diagnosis in their medical record, suggesting the poor awareness that the inpatient community have in recognizing kidney disease [18, 38]. By identifying and informing patients with CKD, a higher awareness of the disease can be obtained, leading to a significant improvement in outcomes [22]. Thus, inpatient screening for impaired eGFR and albuminuria, and education on the importance of detection and early intervention may help identify kidney disease earlier and raise awareness in this setting.

We found that older age was associated with impaired eGFR and albuminuria, this finding is consistent with prior studies [12, 20, 27, 34, 35]. The increased prevalence of kidney disease in the older patients is probably largely as a result of increasing comorbid renal risk factors such as diabetes and hypertension as well as due to structural and functional changes in the aging kidneys [39, 40]. In the present study, 35% of inpatients were more than 60 years of age, and the prevalence of diabetes and hypertension were 31.8% and 29.5%, and 21.7% and 22.5% in patients aged less than 60 years (data not shown). Our results also showed that male gender was associated with increased risk of having albuminuria. Similar results were reported in the Kenyan and Chinese studies [6, 34]. This is, however inconsistent with the results reported in the Southwest Ethiopian study, where male gender was associated with a higher risk of eGFR impairment, but not albuminuria [27]. Therefore, the role of male gender in predicting kidney disease risk warrants further research. Our study has also shown that a family history of kidney disease was associated with impaired eGFR and albuminuria. Most studies show that patients with a family history of kidney disease have an increased risk of impaired eGFR and albuminuria [41] and assessment of kidney disease in subgroups of people with positive family history has been advocated by recent guidelines. However, there are no data that compare differences with a family history of kidney disease in the hospitalized population.

Our results demonstrate diabetes and hypertension as major risk factors for impaired eGFR and albuminuria. Several studies have shown diabetes and hypertension as independent risk factors for kidney disease, as evidenced by impaired eGFR and/or albuminuria [6, 24, 27, 32, 35]. This may reflect that patients with previously known diabetes and hypertension are likely to have higher rates of complications, including renal involvements during hospitalization [42, 43]. Therefore, inpatient screening of these patients for impaired eGFR and albuminuria can be helpful in the early recognition and treatment of kidney disease [44, 45]. In our study, HIV patients are at greater risk of having impaired eGFR and albuminuria. This was consistent with findings from previous study, which revealed that HIV positivity was independently associated with being diagnosed with kidney disease during hospitalization [32]. In the Zambian study by Banda et al, a higher prevalence of renal impairment was found in hospitalized HIV infected patients compared to uninfected patients with a twofold increased risk of developing kidney disease [46]. HIV infection itself, comorbidities and exposure to potentially nephrotoxic antiretroviral agents may play a role in eGFR impairment and albuminuria in HIV/AIDS patients [47].

However, our study experienced several limitations. Given the cross-sectional design, causal relationships between assessed risk factors and renal disease cannot be drawn. For the same reason the diagnosis of CKD was based on a single measurement of serum creatinine and dipstick albuminuria. Thus, there may be a possibility of overestimating the proportion of patients with CKD. However, the exclusion of intensive care unit admissions, patients with possibility of functional proteinuria and patients who had evidence of factors that can cause acute kidney injury or those on medical diagnosis of renal failure should minimize the risk of misclassification of cases as CKD. The MDRD formula used for the analysis has not been validated for use in Ethiopian patients. Further, because the measurement of serum creatinine was not standardized; this hindered us from using the popular Chronic Kidney Disease Epidemiology Collaboration equation.

## Conclusions

In conclusion, CKD is common in adult patients admitted to the internal medicine wards of our hospital in Northeast Ethiopia. About 34% of our patients admitted to the internal medicine wards had CKD according to the diagnosis criterion of impaired eGFR and/or albuminuria, but only 21.6% of affected patients were aware of their condition. These findings highlight the need for feasible approaches to timely identify kidney disease and raise awareness on the importance of early detection and intervention in the inpatient populations. However, the present findings should be confirmed in a larger multicenter study.

## Supporting information

**S1 File.**
(DOCX)

## Acknowledgments

Authors would like to acknowledge all those who agreed to participate in this study, mainly respondents and internal medicine staffs.

## Author Contributions

**Conceptualization:** Temesgen Fiseha, Ermiyas Ahmed.

**Data curation:** Temesgen Fiseha, Ermiyas Ahmed, Semagn Chalie, Angesom Gebreweld.

**Formal analysis:** Temesgen Fiseha.

**Investigation:** Semagn Chalie.

**Methodology:** Temesgen Fiseha, Ermiyas Ahmed, Angesom Gebreweld.

**Software:** Temesgen Fiseha, Ermiyas Ahmed, Semagn Chalie, Angesom Gebreweld.

**Writing – original draft:** Temesgen Fiseha, Ermiyas Ahmed.

**Writing – review & editing:** Temesgen Fiseha, Ermiyas Ahmed, Semagn Chalie, Angesom Gebreweld.

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
