## [Decision Letter · Decision Letter 0]

8 Oct 2020

PONE-D-20-26423

Prevalence and associated factors of impaired renal function and albuminuria among adult patients admitted to a hospital in Northeast Ethiopia

PLOS ONE

Dear Dr. Fiseha,

Thank you for submitting your manuscript to PLOS ONE. After careful consideration, we feel that it has merit but does not fully meet PLOS ONE’s publication criteria as it currently stands. Therefore, we invite you to submit a revised version of the manuscript that addresses the points raised during the review process. Specifically, respond to each of the concerns raised in the accompanying reviewers' comments in addition to the following:

Regarding the eligibility criteria, the text indicated that "Patients were eligible for the study if they were aged 18 years or older, were admitted to the internal medicine wards for at least 48 hours, and had serum creatinine measurements at admission.": (i) were the serum creatinine measurements ordered by the physicians as part of the patients' admission or based on prior investigations and medical records leading to referral? (ii) were the serum creatinine measurements performed by the investigators as part of this study? (iii) When were the blood and urine samples collected for assays from the patients, after 48 hours of being admitted to the internal medicine wards? (iv) The text stated that "A fasting venous blood sample and spot urine specimen were collected from each patient in the morning and then transported to the hospital inpatient laboratory.", explain how patients on hospital admission were made to fast before sample collection, what happened to those that could not fast or where on intravenous fluid or diet? (v) "Blood pressure was measured with a mercury sphygmomanometer after the patients had rested for 5–10 min in the sitting position.", explain whether these measurements were made as part of the admission to the ward or as part of the present study and were they after admission? (vi) Provide description of how patients were screened for "possibility of functional proteinuria and patients who had evidence of factors that can cause acute kidney injury or those on medical diagnosis of renal failure" in this study.

We look forward to receiving your revised manuscript.

Kind regards,

Bamidele O. Tayo

Academic Editor

PLOS ONE

Journal Requirements:

2. Please include additional information regarding the survey or questionnaire used in the study and ensure that you have provided sufficient details that others could replicate the analyses. For instance, if you developed a questionnaire as part of this study and it is not under a copyright more restrictive than CC-BY, please include a copy, in both the original language and English, as Supporting Information. In addition, please include further details concerning the development and validation of this tool.

Reviewers' comments:

Reviewer's Responses to Questions

**Comments to the Author**

1. Is the manuscript technically sound, and do the data support the conclusions?

Reviewer #1: Yes

Reviewer #2: Yes

Reviewer #3: Yes

2. Has the statistical analysis been performed appropriately and rigorously? 

Reviewer #1: Yes

Reviewer #2: I Don't Know

Reviewer #3: Yes

3. Have the authors made all data underlying the findings in their manuscript fully available?

Reviewer #1: Yes

Reviewer #2: Yes

Reviewer #3: Yes

4. Is the manuscript presented in an intelligible fashion and written in standard English?

Reviewer #1: Yes

Reviewer #2: Yes

Reviewer #3: Yes

5. Review Comments to the Author

Reviewer #1: This paper is well written and is relevant. However I have the following comments

Abstract:

Line 38 - 45 (results)

This section can be summarized as follows;

older age (AOR=6.42; 95%CI: 3.36–12.20), a family history of kidney disease (AOR=3.08; 95%CI: 1.39–6.79), diabetes (AOR=2.91; 95%CI: 1.41–6.00), hypertension (AOR=3.83; 95%CI: 1.80–8.18) and HIV (AOR=2.65; 95%CI: 1.15–6.09) were independently associated with both impaired eGFR and albuminuria while male gender (AOR=1.71; 95%CI: 1.02–2.87), was associated with only albuminuria.

Introduction:

Line 75 and 76 (introduction)

This is not entirely true as there is data on CKD among hospitalized patients in Africa. Again, this is not a strong motivation to conduct this research. The author should provide the gap in knowledge clearly, indicate flaws in prior research in Africa and how this current knowledge fills this gap.

Methods:

Line 91 and 92

Were they consecutively recruited and consented?

Please be clear on whether patient provided written informed consent or verbal consent

Line 95

Add the ethical clearance reference number

Line 104 and 105

The mean of the last two readings would have been more accurate as the first reading is often prone to error

Line 110 and 111

Quantification of proteinuria using ACR/UPCR would have been more useful as the dipstick method is prone to problems with dilution and concentration of urine. Further, dipsticks will not detect microalbuminuria. You may add this to your limitations as those with microalbuminuria would have been excluded

Line 112 and 113

Even though, the author excluded patients with suspected AKI based on presence risk factors of AKI or diagnosis of acute renal failure in medical files, this definition is still problematic as there is no baseline creatinine/eGFR. The duration of renal dysfunction or albuminuria; 3 or more or imaging study of shrunken or echogenic kidneys from the clinical files may have strengthened this definition. As it stands some of these patients could have unexplained acute renal failure with no clear risk factors and may have been classified wrongly as CKD in this study.

Line 128

Check this; using a p value of 0.25 as level of significance in the univariate analysis sounds incorrect.

This cannot be right

Results:

Line 140

What cardiovascular diseases were diagnosed? (heart failure, stroke, CAD?)

Discussion:

Line 211 - 213

In this study (Ref 29), did these patients with acute illness have acutely impaired renal function or they were known CKD patients before hospitalization?

If acute, then you cannot compare this with your study cohort as you seem to be studying patients with CKD and not AKI

Line 243

Deleted Disease after CKD

Reviewer #2: Prevalence and associated factors of impaired renal function and albuminuria among adult patients admitted to a hospital in Northeast Ethiopia

REVIEW

Introduction

The authors write on the prevalence of renal impairment, albuminuria and the factors associated with them noting that CKD being increasingly common is often unrecognized in hospitalized patients and that most patients with renal disease in Africa especially, are unaware of their condition. This warranted the study being done to detect this. They noted that the consequences of CKD can be devastating. Therefore, advocating for early detection. The prevalence of renal impairment has been studied in many circumstances and places but has not been studied in their own hospitalized patients and region in east Africa.

The introduction builds a logical case and context for the problem statement which is clear. The research question is implied clearly.

The literature review is up-to-date. The number of references is appropriate and their selection is judicious. The references are mainly primary sources. Ideas are scholarly and acknowledged appropriately and accurately

Comment regarding the novelty and significance of the manuscript.

The research is original, though not novel and addresses important issues which will create avenues for more research on renal diseases and is worth doing. It addresses the need for early evaluation of renal impairment in patients on admission in Africa and it adds to the literature already available on the subject.

Method

The study design is cross sectional and is appropriate for the research question. (prevalence study).

Study population. The setting, locations, and relevant dates, including periods of recruitment, are noted. eligibility criteria, and the sources and methods of selection of participants are noted. The sampling procedures are described as consecutive sampling which is non probabilistic. This study being a local single center, clinic-based study, lacks generalizability which is a limitation of a clinic-based study. Nonetheless, this type of study design is classified as a cross-sectional study.

They did not report numbers of individuals at each stage of study—eg numbers potentially eligible, examined for eligibility, confirmed eligible, included in the study, and analysed. Non-response was not a significant problem in the study.

Threat to internal validity -The manner of selection is non-random, and non-representative which is a source of bias but this is understandable being a small clinic based study. An attempt was made to address potential confounding variables which were excluded in the study.

Instrumentation, Data Collection.

The measurement instruments is appropriate given the study’s major variables; the scoring method is clearly defined for CKD, BP , albuminuria. The terms medical renal failure is rather ambiguous. See footnote below on renal impairment.

They did not specify explicitly which variables were outcome vs exposures. Outcome, exposures, predictors, potential confounders are not clearly stated as such in text but is implied. It is mentioned that observers or raters were trained to take measurements however measures taken to ensure data quality control is not stated.

How the study size was arrived at is appropriate, though formula used to derive it is not referenced.

Data Analysis and Statistics

Statistical tests are simple and appropriate. Data-analysis procedures conform to the research design.

The results are complete, organized and contextualized in a way that is easy to understand. Tables, and figures are used judiciously and agree with the text. Table 1 is presented showing the population characteristics. unadjusted estimates, adjusted estimates and their precision are also presented.

Discussion and Conclusion:

Interpretations of the results are appropriate and alternative interpretations for the findings are considered. Personal perspectives or values related to interpretations are discussed.

guidance for future studies is offered. The study limitations are discussed.

The conclusions key points stand out. My conclusion is consistent with the authors.

Title, authors, and abstract

The title is clear, representative of the content of the study and not misleading. The number of authors is appropriate given the study.

The abstract is complete with essential details presented.

Presentation and Documentation

The text is organized, well written and easy to follow. Reference citations are complete and appropriate.

Scientific Conduct

There are no instances of plagiarism. Ideas and materials of others are correctly attributed.

There is no apparent conflict of interest. There is an explicit statement of approval by an institutional review board (IRB) for the study.

COMMENTS FOR THE AUTHORS

1 Report numbers of individuals at each stage of study—eg numbers potentially eligible, examined for eligibility, confirmed eligible, included in the study, and analysed.

2 Can you clarify what is meant by the term medical renal failure in the patients you excluded. What level of eGFR do you consider renal failure.

3 A statement indicating which of the variables are dependent and independent need to be stated in the analysis plan, say exactly how the prevalence of the condition was derived.

4 The formula used to derive sample size should be referenced.

5 In table 1, Hypertension and diabetes patients do not add up n=370 instead of 369. Reconcile or any explanation? Did you adjust for any other potential confounders in analysis? If yes, make clear which confounders were adjusted for and why they were included.

6 The vocabulary is appropriate, except (in table 1 and any other place with the word illiterate-line 137) use of the term illiterate is inappropriate- change to no formal education.

NOTE: The use of the term renal impairment is ambiguous and generally lacks clarity. The Kidney Disease: Improving Global Outcomes (KDIGO) Consensus Conference held in June 2019, suggested that when referring to ‘decreased or decreasing GFR’, avoid the use of different, poorly defined terms such as ‘impaired kidney function’, ‘renal insufficiency’, ‘renal dysfunction’, ‘renal impairment’, ‘worsening kidney function’ and ‘kidney function decline’. The goal is to facilitate communication within and across disciplines and between practitioners and patients, with the ultimate hope of improving outcomes through consistency and precision. (Refer Andrew S Levey, Kai-Uwe Eckardt, Nijsje M Dorman, Stacy L Christiansen, Michael Cheung, Michel Jadoul, Wolfgang C Winkelmayer, Nomenclature for kidney function and disease: executive summary and glossary from a Kidney Disease: Improving Global Outcomes consensus conference, Nephrology Dialysis Transplantation, Volume 35, Issue 7, July 2020, Pages 1077–1084, https://doi.org/10.1093/ndt/gfaa153

Reviewer #3: This study is pertinent because it highlights an important point that a significant number of patients who are admitted into medical wards but do not have symptoms of kidney disease or require renal replacement therapy may benefit from prevent measures to prevent end-stage kidney disease. Their level of awareness may also be improved when physicians take deliberate steps in this regards.

Anthropometric data are missing and these are important in a study like this in which comprehensive preventive measures are being advocated. Also, could the authors categorize the co-morbidity by age group? In line modern nephrology terminology, 'kidney disease' has generally replaced 'renal disease'.

The last sentence in the discussion (lines 296-298) should be corrected for clarity. Also, the labels on figures 1 and 2 , i.e. 'propertion...' should read 'proportion...'.

6. PLOS authors have the option to publish the peer review history of their article (what does this mean?). If published, this will include your full peer review and any attached files.

Reviewer #1: No

Reviewer #2: No

Reviewer #3: No

---

## [Author Response · Author response to Decision Letter 0]

24 Nov 2020

Response to Academic Editor Comments

Regarding the eligibility criteria, the text indicated that "Patients were eligible for the study if they were aged 18 years or older, were admitted to the internal medicine wards for at least 48 hours, and had serum creatinine measurements at admission.": 

Comment # 1: Were the serum creatinine measurements ordered by the physicians as part of the patients' admission or based on prior investigations and medical records leading to referral?

Response #1: This were the serum creatinine measurements ordered by the physicians as part of the patients' admission and only adults with serum creatinine results available at admission were included in this study.

Comment # 2: Were the serum creatinine measurements performed by the investigators as part of this study?

Response #2: The first serum creatinine during admission was ordered by the physicians as part of the patients' admission and was abstracted from the admission medical records. The serum creatinine measurements for estimating eGFR of each patient was performed by the investigators as part of this study.

Comment # 3: When were the blood and urine samples collected for assays from the patients, after 48 hours of being admitted to the internal medicine wards? 

Response #3: Patients were eligible for the study if they were admitted to the internal medicine wards for at least 48 hours and thus, the blood and urine samples for the study were collected after 48 hours of being admitted to the internal medicine wards.

Comment # 4: The text stated that "A fasting venous blood sample and spot urine specimen were collected from each patient in the morning and then transported to the hospital inpatient laboratory.", explain how patients on hospital admission were made to fast before sample collection, what happened to those that could not fast or where on intravenous fluid or diet? 

Response #4: Patients were excluded from the study if they had been hospitalized in critical condition. After discussing with the attending physician, patients meeting the inclusion criteria were asked to fast before sample collection. Those that could not fast or where on acute intervention were excluded. 

Comment # 5: "Blood pressure was measured with a mercury sphygmomanometer after the patients had rested for 5–10 min in the sitting position.", explain whether these measurements were made as part of the admission to the ward or as part of the present study and were they after admission? 

Response #5: These measurements were made as part of the present study, after 48 hours of being admitted to the internal medicine wards 

Comment # 6: Provide description of how patients were screened for "possibility of functional proteinuria and patients who had evidence of factors that can cause acute kidney injury or those on medical diagnosis of renal failure" in this study.

Response #6: Factors associated with false/transient positive dipstick proteinuria were examined and dipstick proteinuria in subjects with fever (>98.6F), haematuria, leukocyturia, nitrites or very alkaline urine (pH >8.0) were not included into the analysis. To reduce the impact of acute renal failure in the study, increase in serum creatinine by ≥ 0.3mg/dl from admission value (i.e., within 48 hours), and critical care admissions (such as acute circulatory or respiratory failure, presence of infection or cirrhosis) were excluded. We also excluded patients with medical record for renal replacement therapy (dialysis) or those with any mention of acute renal failure, renal failure and contrast administration in the medical files. 

Comment # 1: Please ensure that your manuscript meets PLOS ONE's style requirements, including those for file naming. The PLOS ONE style templates can be found at

Response #2: As suggested, our manuscript follows PLOS ONE formatting to meet PLOS ONE's style requirements

 Comment # 2: Please include additional information regarding the survey or questionnaire used in the study and ensure that you have provided sufficient details that others could replicate the analyses. For instance, if you developed a questionnaire as part of this study and it is not under a copyright more restrictive than CC-BY, please include a copy, in both the original language and English, as Supporting Information. In addition, please include further details concerning the development and validation of this tool.

Response #2: As suggested, the questionnaire for the study was included as Supporting Information. Data collection tool was developed in English with modification from SEEK study. As suggested, it is stated as “Data were collected from patients and their medical records using structured questionnaire which was developed in English with modification from Screening and Early Evaluation of Kidney Disease (SEEK) study (28). The questionnaire was carefully designed and pre-tested on 5% of study population and, based on the results, revision was made to minimize errors. Patients were interviewed to collect data on socio-demographic…” (Line 103-107)

Comment # 3: Your ethics statement should only appear in the Methods section of your manuscript. If your ethics statement is written in any section besides the Methods, please delete it from any other section.

Response #3: As suggested, it is delete and included in the Methods section of the manuscript, stating “The study was approved by the Institutional Review Board of College of Medicine and Health Sciences, Wollo University. Written informed consent was obtained from each study participants after explaining the purpose and procedures of the study. Clinical information obtained in this study was communicated to attending physicians so that they could be used for clinical care.” (Line 97-101)

Response to Reviewer Comments 

Reviewer #1 

This paper is well written and is relevant. However, I have the following comments

Comment # 1: Abstract: Line 38 - 45 (results). This section can be summarized as follows; older age (AOR=6.42; 95%CI: 3.36–12.20), a family history of kidney disease (AOR=3.08; 95%CI: 1.39–6.79), diabetes (AOR=2.91; 95%CI: 1.41–6.00), hypertension (AOR=3.83; 95%CI: 1.80–8.18) and HIV (AOR=2.65; 95%CI: 1.15–6.09) were independently associated with both impaired eGFR and albuminuria while male gender (AOR=1.71; 95%CI: 1.02–2.87) was associated with only albuminuria.

Response #1: If it has to be summarized as suggested above we need to remove the adjusted odds ratios (AOR), as these are odds ratios of albuminuria. As suggested, it is stated as “In multivariable analysis, older age, a family history of kidney disease, diabetes, hypertension and HIV were independently associated with both impaired eGFR and albuminuria while male gender was independently associated with only albuminuria.” (Line 41-44)

Comment # 2: Introduction: Line 75 and 76 (introduction). This is not entirely true as there is data on CKD among hospitalized patients in Africa. Again, this is not a strong motivation to conduct this research. The author should provide the gap in knowledge clearly, indicate flaws in prior research in Africa and how this current knowledge fills this gap.

Response #2: Only 3 published studies are available on CKD and its risk factors among hospitalized patients in Africa. These few studies do not capture the whole picture around the region and this calls for more information, and research should be encouraged to gauge the prevalence of CKD in African countries and define its risk factors. This cross-sectional study was therefore conducted to contribute in filling this knowledge gap. As suggested, it is stated as” However, despite the high prevalence of CKD and its resulting increased in-hospital morbidity and mortality, little is known about the prevalence of CKD in the African hospitalized patients. The few available studies among hospitalized patients in Africa have reported CKD prevalence of 13.5% in Botswana, 38.6% in Kenya (6) and 57.3% in Uganda (27). Understanding the burden and associated risk factors of CKD based on relevant indicators of kidney disease is important for making relevant decisions regarding identification and prevention of the disease in this resource limited region, where access to renal replacement therapy is strictly rationed (25). We therefore ...” (Line 74-81)

Comment # 3: Methods: Line 91 and 92. Were they consecutively recruited and consented? Please be clear on whether patient provided written informed consent or verbal consent

Response #3: We consecutively included inpatients meeting the inclusion criteria until the required sample size was achieved. Written informed consent was obtained from each study participants after explaining the purpose and procedures of the study. This information was mistakenly written in other section besides the Methods, and it now appear in the method section as stated in response # 3 of academic reviewer comments in the additional requirements section.

Comment # 4: Line 95. Add the ethical clearance reference number

Response #4: As suggested, the ethical clearance reference number was added as "..., Wollo University (# 135/13/12).” (Line 98-99)

Comment # 5: Line 104 and 105. The mean of the last two readings would have been more accurate as the first reading is often prone to error

Response #5: BP measurements taken during the first two days of hospitalization were not used for the statistical analyses. Three sequential readings were taken 5 min apart in the sitting position. The subject is asked to relax for 10–15 min (in order to minimize anxiety, which will increase variability). Caffeine was not allowed for at least 1 h and smoking was not allowed for at least 30 min before the BP measurement. If the difference between readings were within the clinically insignificant 0-5 mm Hg range (no error of clinical relevance), the average of the three readings was recorded; otherwise, one additional reading was taken (for 18 patients). The average of three BP measurements was also taken in related African study (in Botswana).

Comment # 6: Line 110 and 111. Quantification of proteinuria using ACR/UPCR would have been more useful as the dipstick method is prone to problems with dilution and concentration of urine. Further, dipsticks will not detect microalbuminuria. You may add this to your limitations as those with microalbuminuria would have been excluded.

Response #6: Albuminuria was evaluated using a semiquantitative urine dipstick test due to unavailability of quantitative albuminuria test. In an effort to correct for problems arising out of variability in urine concentration while screening for albuminuria by dipsticks, the urine specimen was assessed if the urine’s specific gravity was >1.015. It has been also suggested that dipstick testing for albumin, protein, or their ratios to creatinine in hospitalized patients had good or excellent agreement with quantitative methods (Pugia et al. 2001 Albuminuria and proteinuria in hospitalized patients as measured by quantitative and dipstick methods). The albuminuria diagnosis based on a single urinalysis dipstick measurement could lead to overestimating the prevalence of CKD. As suggested, it is stated as “… the diagnosis of CKD was based on a single measurement of serum creatinine and dipstick albuminuria.” in the limitation section (Line 321-322)

Comment # 7: Line 112 and 113. Even though, the author excluded patients with suspected AKI based on presence risk factors of AKI or diagnosis of acute renal failure in medical files, this definition is still problematic as there is no baseline creatinine/eGFR. The duration of renal dysfunction or albuminuria; 3 or more or imaging study of shrunken or echogenic kidneys from the clinical files may have strengthened this definition. As it stands some of these patients could have unexplained acute renal failure with no clear risk factors and may have been classified wrongly as CKD in this study.

Response #7: As already stated in the method section of the manuscript, only adults with serum creatinine results available at admission were included in this study. To reduce the impact of acute renal failure on the study, increase in serum creatinine by ≥ 0.3mg/dl from admission value (i.e., within 48 hours) were excluded. Regarding the duration of renal impairment or albuminuria; based on a single serum creatinine or dipstick albuminuria measurement could lead to overestimating the prevalence of CKD, and this was already stated as the limitation of the study (Line 321-323). Data on imaging study of shrunken or echogenic kidneys from the clinical files was not available because the majority of patients in this study were not hospitalized for kidney disease. As stated in line 323-326, the exclusion of intensive care unit admissions, patients with possibility of functional proteinuria and patients who had evidence of factors that can cause acute kidney injury or those on medical diagnosis of renal failure should minimize the risk of misclassification of cases as CKD in this study.

Comment # 8: Line 128. Check this; using a p value of 0.25 as level of significance in the univariate analysis sounds incorrect. This cannot be right

Response #8: Variable selection should start with the univariate analysis of each variable and variables that show significance (P<0.25) in the univariate analysis should be included in the multivariate analysis (Hosmer et al, 2013. Applied logistic regression). As such, a univariate analysis was done to sort variables candidate for multivariable analyses having value less than 0.25. Multivariable logistic regression analyses were conducted using backward stepwise selection method to identify factors independently associated with impaired renal function or albuminuria. P-value < 0.05 and 95% confidence interval (CI) and AOR was used in judging the statistical significance of the associations between independent variables and the dependent variable.

Comment # 9: Results: Line 140. What cardiovascular diseases were diagnosed? (heart failure, stroke, CAD?)

Response #9: As stated in the method section (Line 109-110), cardiovascular diseases (coronary artery disease, myocardial infarction, heart failure, peripheral vascular disease, and old stroke) were defined as present if recorded as the main admission diagnoses on medical records.

Comment # 10: Discussion: Line 211 – 213. In this study (Ref 29), did these patients with acute illness have acutely impaired renal function or they were known CKD patients before hospitalization? If acute, then you cannot compare this with your study cohort as you seem to be studying patients with CKD and not AKI

Response #10: The lowest creatinine was obtained for each hospital admission and, for up to 3 months before and after the admission. These patients have CKD Stages 3–5 (eGFR < 60 mL/min/1.73 m2).

Comment # 11: Line 243. Deleted Disease after CKD

Response #11: As suggested, the term “Disease” after CKD was deleted (Line 271)

Reviewer #2

Comment # 1: Report numbers of individuals at each stage of study—eg numbers potentially eligible, examined for eligibility, confirmed eligible, included in the study, and analysed.

Response #1: This was a cross-sectional study with only one stage. 

Comment # 2: Can you clarify what is meant by the term medical renal failure in the patients you excluded. What level of eGFR do you consider renal failure.

Response #2: We mean patients with medical diagnosis of renal failure, i.e., renal failure documented in their medical records.

Comment # 3: A statement indicating which of the variables are dependent and independent need to be stated in the analysis plan, say exactly how the prevalence of the condition was derived.

Response #3: As suggested, it is stated as “… 20 software (SPSS Inc., Chicago, IL, USA). We derived means for continuous variables and proportions to describe the characteristics of the study patients as well as the prevalence of impaired renal function, albuminuria and CKD. Comparisons of patients according to the presence of impaired renal function or albuminuria were performed using Chi-square (x2) test and t-test, where appropriate. To determine which factors were associated with the presence of impaired renal function or albuminuria, univariate analysis was conducted with age, sex, residence, education, smoking status, family history of kidney disease, presence of hypertension, diabetes, cardiovascular diseases, respiratory diseases and HIV, and current systolic and diastolic BP as variables. Variables that were found to be significant in univariate analysis (P < 0.25) were included in the multivariable backwards stepwise logistic regression model to identify factors independently associated with impaired renal function or albuminuria. P-value < 0.05 was used to indicate statistical significance.” (Line 133-144)

Comment # 4: The formula used to derive sample size should be referenced.

Response #4: As suggested, it is referenced as “(26)” (Line 96)

Comment # 5: In table 1, Hypertension and diabetes patients do not add up n=370 instead of 369. Reconcile or any explanation? Did you adjust for any other potential confounders in analysis? If yes, make clear which confounders were adjusted for and why they were included.

Response #5: As stated in the method (Line 94-95) and result (Line 145-146) sections of the manuscript, a total of 369 patients who fulfilled the eligibility criteria were consecutively included in this study. Diabetes and non-diabetes/ Hypertension and non-hypertension patients add up n=369 (i.e., we included 369 patients admitted due to a primary diagnosis of diabetes, hypertension, cardiovascular diseases, respiratory diseases, HIV/AIDS and others as recorded on their medical records). To identify factors independently associated with impaired renal function or albuminuria, all variables with univariate P values < 0.25 were included in the multivariable (backward stepwise) logistic regression.

Comment # 6: The vocabulary is appropriate, except (in table 1 and any other place with the word illiterate-line 137) use of the term illiterate is inappropriate- change to no formal education

Response #6: As suggested, the term “illiterate” is changed to “no formal education” (Line 152 and 163 [in table 1])

Reviewer #3

Comment # 1: Anthropometric data are missing and these are important in a study like this in which comprehensive preventive measures are being advocated. Also, could the authors categorize the co-morbidity by age group? In line modern nephrology terminology, 'kidney disease' has generally replaced 'renal disease'.

Response #1: Obtaining anthropometry measurements at admission is challenging, and anthropometric data are frequently documented based on “best-guess” estimates by the clinical care team or parent/relatives, with varying degrees of accuracy. Anthropometric data are not collected during data collection due to fear of procedure associated accidents/discomforts in inpatients, for example patients are expected to stand by themselves, and weight and height are measured in light clothes without footwear. We did not provide data on co-morbidity by age group, but diabetes and hypertension (the two main comorbid renal risk factors) were present in 31.8% and 29.5% of the inpatients aged ≥ 60 years compared to 21.7% and 22.5% of those aged < 60 years. As suggested, it is stated as “In the present study, 35% of inpatients were more than 60 years of age, and the prevalence of diabetes and hypertension were 31.8% and 29.5%, and 21.7% and 22.5% in patients aged less than 60 years (data not shown).” (Line 292-294). As suggested, the term “renal disease” is replaced with “kidney disease” (Line 47, 72, 73, 79, 128, 166, 188, 192, 192, 265, 283, 286, 289, 298, 302, 311, 314, 316 and 334)

Comment # 2: The last sentence in the discussion (lines 296-298) should be corrected for clarity. Also, the labels on figures 1 and 2 , i.e. 'propertion...' should read 'proportion...'.

Response #2: As suggested, it is stated as “HIV infection itself, comorbidities and exposure to potentially nephrotoxic antiretroviral agents may play a role in eGFR impairment and albuminuria in HIV/AIDS patients (46).” (Line 316-318). As suggested, the labels on figures 1 and 2 are corrected and read “proportion (%)”.

Table 2 and 3 were labelled mistakenly as Table 3 and 4. As suggested, the labels are corrected as "Table 2" and "Table 3"

---

## [Decision Letter · Decision Letter 1]

21 Jan 2021

Prevalence and associated factors of impaired renal function and albuminuria among adult patients admitted to a hospital in Northeast Ethiopia

PONE-D-20-26423R1

Dear Dr. Fiseha,

We’re pleased to inform you that your manuscript has been judged scientifically suitable for publication and will be formally accepted for publication once it meets all outstanding technical requirements.

Kind regards,

Bamidele O. Tayo

Academic Editor

PLOS ONE

Additional Editor Comments (optional):

Reviewers' comments:

Reviewer's Responses to Questions

**Comments to the Author**

1. If the authors have adequately addressed your comments raised in a previous round of review and you feel that this manuscript is now acceptable for publication, you may indicate that here to bypass the “Comments to the Author” section, enter your conflict of interest statement in the “Confidential to Editor” section, and submit your "Accept" recommendation.

Reviewer #1: All comments have been addressed

Reviewer #2: All comments have been addressed

Reviewer #3: All comments have been addressed

2. Is the manuscript technically sound, and do the data support the conclusions?

Reviewer #1: Yes

Reviewer #2: Yes

Reviewer #3: Yes

3. Has the statistical analysis been performed appropriately and rigorously? 

Reviewer #1: I Don't Know

Reviewer #2: Yes

Reviewer #3: Yes

4. Have the authors made all data underlying the findings in their manuscript fully available?

Reviewer #1: Yes

Reviewer #2: Yes

Reviewer #3: Yes

5. Is the manuscript presented in an intelligible fashion and written in standard English?

Reviewer #1: Yes

Reviewer #2: Yes

Reviewer #3: Yes

6. Review Comments to the Author

Reviewer #1: All comments addressed. However, for choosing P < 0.25 for the univariate analysis as explained by the author, it will be appreciable if the author can check for co-linearity between the dependent and independent variables.

Reviewer #2: (No Response)

Reviewer #3: The authors have provided adequate answers to questions raised in the first round of review. I have also read their responses to other reviewers' comments. Within the limitations so stated in a cross-sectional section the manuscript has improved. Again, this paper draws attention to the burden of CKD among hospitalised patients in the medical wards, many of whom have never had opportunity of visiting family physicians to screen for CKD. CKD is in the community as well as in the hospital, and patients on our beds should not go undetected.

7. PLOS authors have the option to publish the peer review history of their article (what does this mean?). If published, this will include your full peer review and any attached files.

Reviewer #1: No

Reviewer #2: No

Reviewer #3: No

---

## [Editor Report · Acceptance letter]

25 Jan 2021

PONE-D-20-26423R1 

Prevalence and associated factors of impaired renal function and albuminuria among adult patients admitted to a hospital in Northeast Ethiopia 

Dear Dr. Fiseha:

I'm pleased to inform you that your manuscript has been deemed suitable for publication in PLOS ONE. Congratulations! Your manuscript is now with our production department. 

Kind regards, 

on behalf of

Dr. Bamidele O. Tayo 

Academic Editor

PLOS ONE